# The Formation of the Epiphyseal Bone Plate Occurs via Combined Endochondral and Intramembranous-Like Ossification

**DOI:** 10.3390/ijms22020900

**Published:** 2021-01-18

**Authors:** Ángela Fernández-Iglesias, Rocío Fuente, Helena Gil-Peña, Laura Alonso-Durán, Fernando Santos, José Manuel López

**Affiliations:** 1Division of Pediatrics, Department of Medicine, Faculty of Medicine, University of Oviedo, 33006 Oviedo, Asturias, Spain; angelafiglesias@gmail.com (Á.F.-I.); rociofuenteperez@gmail.com (R.F.); hgilpena@gmail.com (H.G.-P.); laurita.alonso86@gmail.com (L.A.-D.); fsantos@uniovi.es (F.S.); 2Instituto de Investigación Sanitaria del Principado de Asturias (ISPA), 33011 Oviedo, Asturias, Spain; 3Department of Pediatrics, Hospital Universitario Central de Asturias (HUCA), 33011 Oviedo, Asturias, Spain; 4Department of Morphology and Cellular Biology, Faculty of Medicine, University of Oviedo, 33006 Oviedo, Asturias, Spain

**Keywords:** bone plate, endochondral ossification, intramembranous ossification, chondrocyte, osteoblast

## Abstract

The formation of the epiphyseal bone plate, the flat bony structure that provides strength and firmness to the growth plate cartilage, was studied in the present study by using light, confocal, and scanning electron microscopy. Results obtained evidenced that this bone tissue is generated by the replacement of the lower portion of the epiphyseal cartilage. However, this process differs considerably from the usual bone tissue formation through endochondral ossification. Osteoblasts deposit bone matrix on remnants of mineralized cartilage matrix that serve as a scaffold, but also on non-mineralized cartilage surfaces and as well as within the perivascular space. These processes occur simultaneously at sites located close to each other, so that, a core of the sheet of bone is established very quickly. Subsequently, thickening and reshaping occurs by appositional growth to generate a dense parallel-fibered bone structurally intermediate between woven and lamellar bone. All these processes occur in close relationship with a cartilage but most of the bone tissue is generated in a manner that may be considered as intramembranous-like. Overall, the findings here reported provide for the first time an accurate description of the tissues and events involved in the formation of the epiphyseal bone plate and gives insight into the complex cellular events underlying bone formation at different sites on the skeleton.

## 1. Introduction

The epiphyseal bone plate is a flat bony structure located between the epiphysis and the metaphysis of the long bones [1]. It holds the growth plate cartilage, providing the weakest area of the growing bone with strength and stability. In addition to structural support, the epiphyseal bone plate also has a nutritional role by allowing the passage of blood vessels from the epiphysis to form a capillary network that supplies the growth cartilage with oxygen, nutrients, and chemical signaling. 

The epiphyseal bone plate is formed by ossification of the lower part of the epiphyseal cartilage after the development of the secondary ossification center. At the beginning, the secondary ossification center expands in all directions to form spongy bone trabeculae with a radial orientation. However, the mode of ossification substantially changes when the front nears the region facing the resting cartilage of the growth plate. At this location, cartilage is directly transformed into compact bone tissue to generate a flat plate with several layers of densely packed bony lamellae oriented transverse to the long axis of the bone [2]. The formation of the epiphyseal bone plate has some peculiar characteristics that make it different from a standard endochondral ossification process. Firstly, it is considered as a specialized extension of the cortical bone [3], which is usually formed through direct-intramembranous ossification. Secondly, it is oriented perpendicular to the long axis of the bone, an unusual orientation since most bone structures are oriented such that they are stronger in the direction in which they support load [4]. Thirdly, it develops very quickly, so that transformation of cartilage into bone occurs faster than anywhere else in the bone [5]. Finally, it is formed in the region immediately adjacent to the resting cartilage of the growth plate and this close spatial proximity suggests that the latter could exert a great deal of influence on its development [6].

Despite the above considerations and the fact that the epiphyseal bone plate could be considered a structural component of the growth plate that influences its function, it has been largely unexplored and is one of the least understood parts of the long bones. Even little is known about how it is affected by diseases associated with growth failure and bone abnormalities. On this basis, it was the aim of the present work to obtain an overall overview of how the epiphyseal bone plate is formed in physiological conditions. To this end, we have performed a time course analysis of the epiphyseal development utilizing tibias of rats at different stages of growth. We have used light, confocal, and scanning electron microscopy to analyze the cellular aspects of the tissue development. The study provides a framework for future research on pathological conditions.

## 2. Results

The formation of the epiphyseal bone plate is a continuous process but in order to make the description easier to follow, we divided it into four stages characterized by changes in the amount of cartilage and bone formation.

Stage 1: The beginning of epiphyseal bone formation. The proximal epiphysis of the rat tibia was entirely cartilaginous at 3 days of age (Figure 1A). Despite its relatively homogeneous appearance, three zones without clearly defined borders could be recognized in the cartilage: (1) An outer cap of flatter chondrocytes located at the epiphyseal end; (2) an extensive intermediate zone with oval chondrocytes oriented randomly; and (3) a deep zone with chondrocytes aligned in columns. Zones 1 and 3 corresponded to the prospective articular and metaphyseal growth plate cartilages, respectively, whereas zone 2 corresponded to the epiphyseal cartilage that specifically guides bone tissue expansion to establish the secondary ossification center. The first cartilage canals in the epiphysis were observed at the age of 5 days, and by the day 8, chondrocytes in the middle of the epiphysis became hypertrophic, canals spread into the central region and the secondary ossification center was established (Figure 1B). The secondary ossification center was firstly spherical and smoothly bordered by hypertrophic cartilage (Figure 1C), but by 14 days of age, it became increasingly irregular and presented a heterogeneous inner surface with both smooth and irregular areas (Figure 1D). 

Smooth and irregular areas in the secondary ossification center were not randomly distributed; smooth inner surfaces were found at the upper and lateral portions of the epiphysis, within a semicircular region below the articular cartilage, whereas a rugged inner surface was observed in the lower portion, that overlying the metaphyseal growth plate cartilage (Figure 1E). The upper-lateral region had the spatial arrangement of an inverted U-shape that underlay the surface of the bone head. It was relatively wide and uniform in thickness, although interruptions due to the primary channels derived from the perichondrium could be found (Figure 1D). The size and shape of chondrocytes varied along the radial direction and they were replaced by trabecular bole at the border with the bone marrow, so that the process of ossification in this cartilage was typically endochondral.

The lower region of the epiphyseal cartilage presented an irregular outline, with some tongues or finger-like extensions of marrow tissue that deeply penetrate into the epiphyseal cartilage to reach the border of the growth plate cartilage (Figure 1E). Local thinned areas of the cartilage resulting from marrow tissue extension appeared as holes or depressions that could be delineated by the very low or absent expression of type X collagen (Figure 1F) and high alkaline phosphatase (AP) staining (Figure 1G). Marrow tissue entering the cartilage contained blood vessels, numerous intervascular mesenchymal-type cells and some large multinucleated TRAP positive cells (Figure 1H). The cartilage-marrow interface did not have a uniform appearance since the stage of maturation of terminal chondrocytes varied in different areas depending on the thickness of the cartilage below. Unthinned areas presented terminal chondrocytes showing a hypertrophic-like shape whereas terminal chondrocytes at thinned areas were smaller, flatter, and stained slightly darker (Figure 1I). There was not a proper ossification front with invading capillary sprouts entering in open empty chondrocytic lacunae since most terminal lacunae remained closed with chondrocytes showing no signs of degeneration (Figure 1I). Mineral deposition occurred in the interterritorial matrix. Early scattered focal mineral deposits coalesced to form larger mineral aggregates in the matrix of terminal chondrocytes. Compact mineral aggregates were patchy and distributed towards the cartilage border, without a uniform mineralization front (Figure 1I). No mineralization of the cartilage matrix was detectable in areas of the cartilage thinned by the extension of the marrow tissue, where terminal chondrocytes were smaller, flatter and had not alkaline phosphatase activity (Figure 1I).Von Kossa positive staining was observed in small clusters of cells located in the intervascular spaces at the mid-depth zones of the finger-like extensions of marrow tissue (Figure 1I). These cells stained with alkaline phosphatase and were positive for osteocalcin expression (Figure 1J,K). They were small in size and appeared surrounded by a narrow mineralized matrix (Figure 1I,L). Formation of mineralized matrix was observed among cells both in the periphery and the center of clusters. These mineralized areas were located adjacent to capillaries and had no apparent connection with the cartilage surface; no remnants of mineralized cartilage matrix that could serve as a scaffold for bone deposition were found (Figure 1I,L). Capillaries occupied the deepest part of the tissue extensions and were mainly sinusoidal in shape with moderately dilated lumina. They were close to the cartilage, with intimate juxtaposition of the endothelial cells to the cartilaginous matrix (Figure 1I,L). The zones of contact presented a fairly smooth outline and were outlined by a thin band of less toluidine blue stained cartilaginous matrix, a fact indicating proteoglycan lost (Figure 1M). Analysis of 1 μm semithin sections revealed the presence of large, rounded-shaped cells with a centrally placed spherical nucleus and pale-staining cytoplasm surrounded by a thin rim of metachromatic substance (Figure 1M). These cells were interspersed among smaller cells in the intervascular space, close to mineralizing areas, and expressed type II collagen (Figure 1N). and could be considered as chondrocytes freed form the cartilage matrix that retained part of the proteoglycans of the pericellular matrix. 

Stage 2: The earliest recognizable bone plate. The epiphyseal cartilage was progressively replaced by bone between postnatal day 11 and 14 so that by day 19 the epiphysis exhibited greatly reduced cartilage and a well-formed marrow space with a large amount of newly formed bone tissue (Figure 2A). 

Marked differences in bone microarchitecture were observed between the upper and the lower borders of the epiphyseal bone marrow cavity at this age. Bone tissue at the upper portion consisted of narrow trabeculae that were continuous with the radially oriented calcified longitudinal septa of the hypertrophic cartilage. The overall radial organization of the trabeculae at this region gave the bone marrow a “bicycle wheel” pattern (Figure 2A). On the other hand, bone at the lower portion had a bridge-like structure with a series of “piers”, which corresponded to the areas of direct contact between the cartilage and the bone, spaced from each other by plate-like bony segments of moderate thickness separated from the underlying epiphyseal cartilage by a tissue composed of capillaries and intervascular cells (Figure 2A). The upper and lower surfaces of the mineralized plate-like elements differed in the shape and arrangement of the adjoining cells; the lower surface was covered by several layers of spindle-shaped cells arranged in diffuse sheets whereas the upper surface was lined by a layer of cells with a more regular shape (Figure 2C,D). Some of the cells closest to the lower bone surface became partially or completely embedded in the mineralized matrix whereas the farthest ones were close to capillaries (Figure 2C,D). Cells at the upper side were more cuboidal and displayed a pseudoepithelial arrangement (Figure 2C,D), characteristic features of active osteoblasts lining a bone forming surface. Capillaries were moderately dilated and lied in close contact with the cartilage surface (Figure 2D,E).There was little to no positive immunohistochemical staining for CD31 in epiphyseal capillaries unlike those at the border with the hypertrophic chondrocytes in the metaphysis that were positively immunostained (Figure 2F).

The “piers” of the bridge corresponded to areas of contact between cartilage and the mineralized tissue, where the two tissues laid side-by-side without any interposing transition region (Figure 2G,H). The two tissues were stained differentially with Toluidine blue in semithin sections (Figure 2G) and also with Alcian blue/alizarin red in paraffin sections (Figure 2H) and a sharp boundary was visible between them. Likewise, von Kossa staining also revealed a clear border between mineralized and non-mineralized tissues (Figure 2I). The cartilage appeared to be covered by several layers of flat spindle-shaped cells embedded in mineralized matrix. These cells were arranged in rows with their long axis oriented parallel to each other and the surface of the tissue and the shape of the cells on the surface varied from cuboidal to flat (Figure 2G). Morphological features were compatible with laminar bone deposited directly on the cartilage, with cuboidal osteoblasts in active bone forming areas and quiescent bone lining cells in not active zones. Chondrocytes directly covered by the calcified tissue had not the appearance of hypertrophic chondrocytes, they were not substantially enlarged and had a well-preserved structure, with moderate abundant cytoplasm and clearly visible nuclei containing nucleoli (Figure 2G). 

Stage 3: Thickening by appositional growth. An almost continuous bony structure is found along the border of the growth plate cartilage by the age of 24-days. Analysis of calcein labeling under the confocal microscopy evidenced that fluorescence was located mainly around the blood vessels, indicating that bone tissue formation was associated with the pattern of vascularization (Figure 3A). Osteoblasts and osteoclasts were located on the opposite sides of the bone plate in a manner consistent with a bone-forming surface in the side oriented toward the bone marrow and a bone resorption surface in the side oriented toward the cartilage of the growth plate (Figure 3B–J). 

Osteoblasts formed a continuous epithelium-like monolayer of cuboidal cells on the surface of the upper side. Directly underneath the layer of osteoblasts was a thick layer of osteoid and beneath the latter was the mineralized bone. Osteoid and mineralized bone could be distinguished in semithin sections by the positive toluidine blue staining of the former and the almost complete lack of staining of the latter, so that a sharp boundary between the unmineralized and mineralized zones was observed (Figure 3B,C). Microscopic analysis in semithin sections provided a clear picture of a gradual addition of new osteoid and osteoid-osteocytes from the surface to the deeper layers. The osteoid-osteocytes nearest the bone-forming surface stained intensely with toluidine blue while those further away were smaller, flattened and stained moderately (Figure 3C). Osteocalcin expression was visualized as a smoothly continuous band on the surface of the upper side whereas it was only found in a few scattered cells at the lower side (Figure 3D,E). On the other hand, large multinucleated TRAP-positive osteoclasts (Figure 3F–H) and high gelatinase B expression were observed at the lower side (Figure 3I,J), both indicating active osteoclastic bone resorption at this area. 

Stage 4: Mature bone plate. The cartilage of the lower region of the epiphysis was completely removed and the epiphyseal bone plate acquired most of its adult structural characteristics By the age of 35 days. It appeared as a flat layer of bone tissue separated from the underlying growth plate cartilage by a gap about 50 μm wide, although the two structures remained bridged by a discrete number of thin bony extensions where a direct contact between cartilage and mineralized bone persisted (Figure 4A,B). Several relatively thick trabeculae arose from the upper side of the bone plate and joined it to the trabecular meshwork of the epiphysis (Figure 4A,C). The plate was perforated by tiny foramina (Figure 4D) through which blood vessels passed from the epiphyseal marrow to ramify and form a rich network close to the resting cartilage. The bone tissue had a layered arrangement, but a preferential fiber orientation was not systematically present. Collagen fibers were wavy and appeared highly interlaced (Figure 4E). Likewise, the osteocytes were not aligned in clear layers (Figure 4B,E). The bone tissue in the bone plate appeared to be structurally intermediate between woven-fibered and parallel-fibered bone and, then it could be categorized as parallel-fibered bone.

## 3. Discussion

The results in the present paper give, for the first time, the microscopic details of the formation of the epiphyseal bone plate. This process involves a mode of ossification that is temporally, spatially, and histologically distinct from other bony structures. During the formation of the epiphyseal bone plate, bone is deposited by osteoblasts on remnants of mineralized cartilage matrix that serve as a scaffold but also on non-mineralized cartilage surfaces and as well as within the perivascular space. The process occurs simultaneously at various sites of the epiphyseal cartilage located close to each other within a short period of time, so that a core of the sheet of bone is established very quickly. Once this early bone plate is formed, thickening and reshaping takes place through appositional growth, bone is laid down by osteoblasts in one side whereas osteoclastic resorption takes part on the opposite side. The final result is the fast formation of dense parallel-fibered bone, structurally intermediate between woven bone and lamellar bone. Time is critical in this process because the structure must be formed to reinforce the growth plate cartilage before the increase in the mechanical load associated with the onset of walking. Together, our results show that the lower portion of the epiphyseal cartilage quickly completes its transformation into dense bone in a manner that is different to any other site in the bone and may be interpreted as mainly intramembranous-like in type, although it takes place in close relationship with a cartilage. 

Intramembranous and endochondral ossification have been usually regarded as separate and different modes of ossification. Nevertheless, osteoblasts are the cells responsible for bone formation in both types and the main difference between them rests on whether osteoblasts deposit bone matrix with or without a preformed cartilage model. Osteoblasts are not able to deposit bone matrix by themselves in one specific direction since they have not different domains of the plasma membrane. Osteoblasts require a spatial framework serving as a scaffold to migrate and deposit properly structured bone tissue, so that each type of scaffolds greatly influences the osteoblast behavior [7]. Bone formation occurs on either one of three types of scaffolds: Unmineralized tissue, mineralized cartilage, and preexisting bone. The first corresponds to de novo intramembranous bone formation, the second to endochondral bone formation and the third to periosteal bone formation. The findings of our study support that the formation of the epiphyseal bone plate concurrently involves these three types of scaffolds for osteoblast activity and this result is in accordance with previous studies assessing that both intramembranous and endochondral ossification occur simultaneously during the development of some types of bones. In this way, although it is generally assumed that a long bone is formed by endochondral ossification, much of its postnatal ossification occurs subperiosteally and so is essentially intramembranous [8]. Additionally, a combination of both intramembranous and endochondral ossification has been reported to occur simultaneously during craniofacial bone development [9], clavicle formation [10], fracture healing of mandible [11], and bone lengthening through callus distraction [12,13]. 

The development of long bones is a complex process that results in the formation of a mature form with a specificarchitectural organization that is of major importance because the strength of a bone is partially dependent of the occurrence of accurate amounts of compact and cancellous osseous tissues arranged in a precise spatial pattern [14,15]. Although bone formation has been deeply studied at the metaphyseal side of the growth plate [16,17,18,19], the amount of research on epiphyseal ossification is so scarce that there is not even consensus on the terminology of the tissue that forms the epiphyses at early stages of the development, when they are entirely cartilaginous. The name “epiphyseal cartilage” has been used by several authors [20,21,22] but it has never been accepted into general use since it may lead to a certain ambiguity because other authors have used it as a synonym of growth plate. Since there is no available alternative name, in the present work we use the term “epiphyseal cartilage” to refer the cartilage from which epiphysis are entirely built at perinatal stages. This cartilage later will give rise to the articular cartilage at the end of the bone, trabecular bone in the middle and a plate of dense bone at the base of the epiphysis. The present work shows that the formation of the secondary ossification center separates a previous continuous tissue bulk into upper and lower parts that will evolve in different ways. The upper cartilage gives rise to the articular cartilage and to the secondary growth plate and whereas the lower one generates the bone plate.

The secondary growth plate is a hemispherical structure that determines the size and three-dimensional shape of the epiphysis in each particular bone [4,23]. Ossification at the secondary growth plate presents many comparable aspects to that at the metaphyseal growth plate. Chondrocytes first proliferate at the upper portion and subsequently undergo an increase in cell size. Mineralization occurs in the radial septa between the terminal hypertrophic chondrocytes and osteoblasts form bone tissue on persisting calcified radial septa that serve as scaffolding according to a typical endochondral ossification. By contrasts, the ossification of the lower epiphyseal cartilage varies substantially from a typical endochondral ossification. Chondrocytes do not undergo hypertrophy and do not produce a mineralized matrix that guides bone deposition by osteoblasts. Furthermore, the vascular invading front is not smooth but there are sites with increased cartilage resorption that give rise to finger-like extensions of the bone marrow tissue that penetrate the cartilage deep enough to reach the resting zone of the growth plate. These structures have not been explicitly reported in previous studies but can be observed in the iconographic content of some of these preceding works [4,24,25,26,27,28,29]. Their formation involves cartilage resorption that is uncoupled to bone formation, a feature also found in the development of the cartilage canals that precede the formation of secondary ossification center. Cartilage canals are channels containing blood vessels and perivascular cells that originate at the perichondrial surface and invade the epiphyseal cartilage in a radial direction [24,30]. They serve to nourish the cartilage and also as a path to supply the epiphysis with mesenchymal stem cells coming from the perichondrium [31,32], two functions that could be also performed by the channels through lower epiphyseal cartilage. The mature bone plate is perforated and this is of major importance because these perforations allow the passage of blood vessels from the epiphysis to form a capillary network close to the resting cartilage of the growth plate. Blood vessels penetrate the epiphyseal cartilage before its ossification and the perforations are formed by subsequent layering of bone tissue around them. The extensions of the bone marrow tissue into the lower part of the epiphyseal cartilage could be a path by which osteogenic cells could quickly reach the zone where the bone plate is going to be formed. It is interesting to note that the osteogenic cells could reach different depths into the partially degrading cartilage. Since the microenvironmental conditions at different depths vary because of the different stage of maturation of terminal chondrocytes and also of the greater or less proximity to the resting cartilage of the growth plate, osteogenic cells can respond differently depending on their location. This could explain the findings in our study that osteoblasts may use different types of scaffolds for bone deposition during the formation of the epiphyseal bone plate. In this way, it is well known that osteoblasts can respond differently to stimuli depending on the bone compartment where they are located [33]. It has been reported that osteoblasts at the metaphyseal side of the growth plate radically change the mode of bone deposition to form a proper horizontal bone plate after surgical excision of the growth plate and subsequent reimplantation in the inverted orientation [34,35]. 

The results in our study indicate that capillaries are located just ahead of the finger-like extensions of bone marrow tissue that penetrate into the epiphyseal cartilage before the formation of the bone plate. Furthermore, direct bone formation occurs in the interstitial space close to them, a fact that suggests that capillaries may be part of the scaffold that guides the deposition of bone matrix by osteoblasts. This result is coherent with the general finding in any type of ossification that bone formation is tightly associated with the development of new blood vessels. Endothelial cells produce paracrine factors that affect osteoblast function or differentiation [36,37]. A specific subtype of capillary termed type H has been recently reported to be specifically associated with osteogenesis [38,39]. These capillaries are densely surrounded by osteoprogenitor cells and characterized by high expression of CD31 and endomucin. Type H vessels have been studied in trabecular bone adjacent to the growth plate, where they are organized as straight columns interconnected at their distal end and can direct bone formation by releasing factors that stimulate proliferation and differentiation of osteoprogenitor cells into osteoblasts. In the present study, we have found a close relationship between capillaries and bone deposition in the interstitial space during the formation of the bone plate, but their structural characteristics and the lack of immunoreactivity for CD31do not match with those of the metaphyseal type H vessels. 

One remarkable finding of the present study is the existence of chondrocytes freed from the cartilage matrix in the zone where the bone plate is formed. Released viable chondrocytes have also been reported during the formation of cartilage canals, at the beginning of the formation of the secondary ossification centers [32,40] but, to our knowledge, they have not been reported before in later stages. Freed chondrocytes have been suggested to re-enter the cell cycle and further differentiate to other cell types, including osteoblasts, and take part in the ossification of the epiphysis [41]. Consistent with this possibility is the fact that chondrocytes display a remarkable capacity to differentiate into other cell types depending on the characteristics of the local microenvironment [42]. Furthermore, it has been reported that the initiation of endochondral ossification at the secondary ossification center occurs via the direct transdifferentiation of chondrocytes to osteoblasts and not by canonical endochondral ossification processes [43]. Likewise, chondrocytes located at singular sites like the border of the cartilage rudiments and the bony collar [44], the closing growth plate of aged rats [29], or the mixed spicules during fracture healing [45] differentiate according to uncommon ways and gives rise to exceptional phenotypes [46]. Even the long time accepted idea that hypertrophic chondrocytes are programmed to die during the process of endochondral ossification has been revised by recent studies using lineage tracing techniques that have identified different models for chondrocyte-to-osteoblast transdifferentiation [33,47,48,49]. In this way, results in the present study support the possibility that chondrocytes may be contribute to the formation of the epiphyseal bone plate. 

In summary, our present study provides an accurate description of the tissues and events involved in the formation of the epiphyseal bone plate. We have identified some features that have not been previously reported and may be significant for understanding this bone structure. Overall, our work gives insight into the complex cellular events underlying bone formation at different sites on the skeleton and provides a basis for future studies on this relatively little investigated region of the bone.

## 4. Materials and Methods 

Male Sprague-Dawley rats were obtained from the animal facility building of the University of Oviedo. Rats were sacrificed on days 3, 5, 8, 11, 14, 19, 24, and 35 after birth. We used five animals per age group except in the age-group of 35 days in which an additional rat was included (*n* = 6). Rats were injected intraperitoneally with calcein 20 mg/kg (Sigma, St Louis, MO, USA) three days before sacrifice. The procedures involving animals and their care were conducted according to Spanish law on the use of experimental animals, which acknowledges the European Directive 86/609. The project proposal was approved by the Ethical Committee of the University of Oviedo, Spain (14 July 2015, PROAE 19/2015).

Tibiae were isolated and cut through the sagittal plane of the proximal epiphysis into two equal sized parts, obtaining four tibial halves from each animal. One tibial half was embedded in paraffin to obtain sections which were used for both histochemical and in situ hybridization studies. The second tibial half was processed for mineralization studies. The third tibial half was processed to obtain semithin sections and the last tibial half was processed for confocal microscopy. Tibiae of the additional rat of the 35 days old group were processed for scanning electron microscopy. 

For histochemical, immunohistochemistry and in situ hybridization studies, tissues were fixed by immersion in 4% paraformaldehyde at 4 °C for 12 h, rinsed in PBS, decalcified in 10% EDTA (pH 7.0) for 48 h at 4 °C, dehydrated through a graded ethanol series, cleared in xylene and embedded in paraffin. Sections were serially cut at a thickness of 5 µm and mounted on Superfrost Plus slides (Menzel-Glaser, Thermo Fisher Scientific, Germany). Trichromic staining was carried out by using Weigert’s hematoxylin/Alcian blue/picrofuchsin, which distinguished cartilage matrix (blue) from bone matrix (red). For analysis of alkaline phosphatase (AP) staining, sections were pretreated with 10 mM MgCl_2_ solution to reactivate the enzyme and subsequently incubated with a substrate solution containing 0.16 mg/mL 5 bromo 4 chloro 3 indolylphosphate and 0.33 mg/mL nitroblue tetrazolium in 100 mM Tris, 100 mM NaCl, 50 mM MgCl_2_, pH 9.5, for 30 min at room temperature. Tartrate resistant acid phosphatase (TRAP) activity, a marker of osteoclast precursor cells and mature osteoclasts, was also determined histochemically by incubation of sections with 50 mM sodium acetate (pH 5.2) containing 0.15% Napthol AS TR phosphate, 50 mM sodium tartrate, and 0.1% Fast Red T.R. (Sigma).

Immunohistochemistry for CD31 was applied by using a rabbit polyclonal antibody (Abcam, Cambridge, MA, USA). Sections were deparaffinized, hydrated and treated with a solution of 0.05% pepsin (Sigma) in 0.01 M HCl (pH 2) at 37 °C for 15 min for antigen retrieval. After H_2_O_2_ inactivation and unspecific antibodies blocking, slices were incubated overnight with the primary antibody (1:100). Immunostaining was then performed by using an ExtrAvidin Peroxidase Staining Kit for rabbit antibodies (Sigma-Aldrich, Saint Louis, MO, USA). 

In situ hybridization studies were performed as reported previously [50]. Sections were hybridized to ^35^S-labeled antisense riboprobes and subsequently exposed to photographic emulsion at 4 °C for several days, developed, fixed, cleared and counterstained with 0.02% toluidine blue. Sections hybridized with a labeled-sense riboprobe were used as negative controls. Either, sense or antisense ^35^S-uridine triphosphate-labeled RNA probes were synthesized from the corresponding linearized DNA using the appropiate RNA polymerases. Probes for in situ hybridization were as follows: The probe for type II collagen was a 550 bp Pst I fragment from the amino-terminal portion of rat pro-α1(II) chain cloned in PGEM 3Zf-vector. Type X collagen probe was a 650-bp HindIII fragment containing 360 bp of non-collagenous (NC1) domain and 290 bp of 3′-untranslated sequence of the mouse type X collagen gene, subcloned into the HindIII site of pBluescript. The rat and mouse genes (RGD ID: 2371 vs. RGD ID: 735282) share 96.0–97.0 sequence similarity (ncbi.nlm.nih.gov/genome). The gelatinase B probe was generated by subcloning into pBluescript a 1353 BamHI fragment obtained by RT-PCR from embryo RNA with the following oligonucleotides: 5′ TGGCACCATCATAACATCACCT and 5′ AGAAGAAAATCTTCTTGGGCTG (GenBank acces. nb. NM_031055). The probe for murine osteocalcin consisted of a 212-bp fragment amplified from embryo RNA with the oligonucleotides 5′-TCTCTCTGCTCACTCTGCTGG and 5′AGCAGGGTTAAGCTCACACTand subcloned into pBluescript vector. Digoxigenin-11-UTP-labeled single-stranded RNA probes were also used for type-II collagen to get a better spatial resolution of chondrocytes freed form the cartilage matrix. Probes were prepared with the DIG RNA labeling mix (Boehringer Mannheim, IN, USA), and the hybridized probe was detected with the alkaline-phosphatase-coupled anti-DIG antibody (Boehringer Mannheim).

For studies of matrix mineralization, tissues were fixed in 4% paraformaldehyde in PBS for 5 h at 4 °C, dehydrated in acetone, embedded in Durkupan-ACM (Sigma) and sectioned at 2 μm on a Reicher Ultracut E ultramicrotome. The von Kossa staining was used to detect mineralization by setting sections in 1% AgNO_3_ for 60 min at room temperature and fixed with 5% sodium hyposulfite. 

Semithin sections were obtained from tissues fixed in 2% glutaraldehyde and 0.7% ruthenium hexamine trichloride (RHT) (Strem Chemicals, MA, USA) in 0.05 M cacodylate buffer, pH 7.4, for 3 h at 4 °C. They were then postfixed in 1% osmium tetroxide and 0.7% RHT in cacodylate buffer for 2 h. After washing, they were dehydrated with acetone, embedded in Durkupan-ACM (Sigma), sectioned at 0.5 μm on a Reicher Ultracut E ultramicrotome and stained with toluidine blue. 

Bone samples for confocal microscopy were processed as previously reported [51]. Briefly, tibial samples were fixated in glutaraldehyde 2%, RHT 0.5%, and calcium chloride 5 mM in 0.025 M sodium cacodylate buffer (pH 7.4, osmolarity 300 mOsm). Then, they were embedded in epoxy resin Durcupan (Sigma), thinned by grinding to obtain 100-µm thick bone sections and imaged with a confocal microscope Leica TCS SP8 (Leica Microsystems, Wetzlar, Germany) equipped with a pulsed white light laser (470–670 nm), Acousto-Optical Beam Splitter (AOBS), and two internal hybrid single photon counting detectors, which was operated by Leica Application Suite X program (Leica Microsystems).

Bone samples for scanning electron microscopy were dehydrated in a graded series of acetone and critical-point dried in a Bomar SPC-900 EX critical-point dryer. Samples were glued to standard SEM stubs, coated with gold in a Sputtering Balzers SCD 004 and viewed with a JEOL JSM-5600 scanning electron microscope operated at 20 kV. 

## Figures and Tables

**Figure 1 ijms-22-00900-f001:**
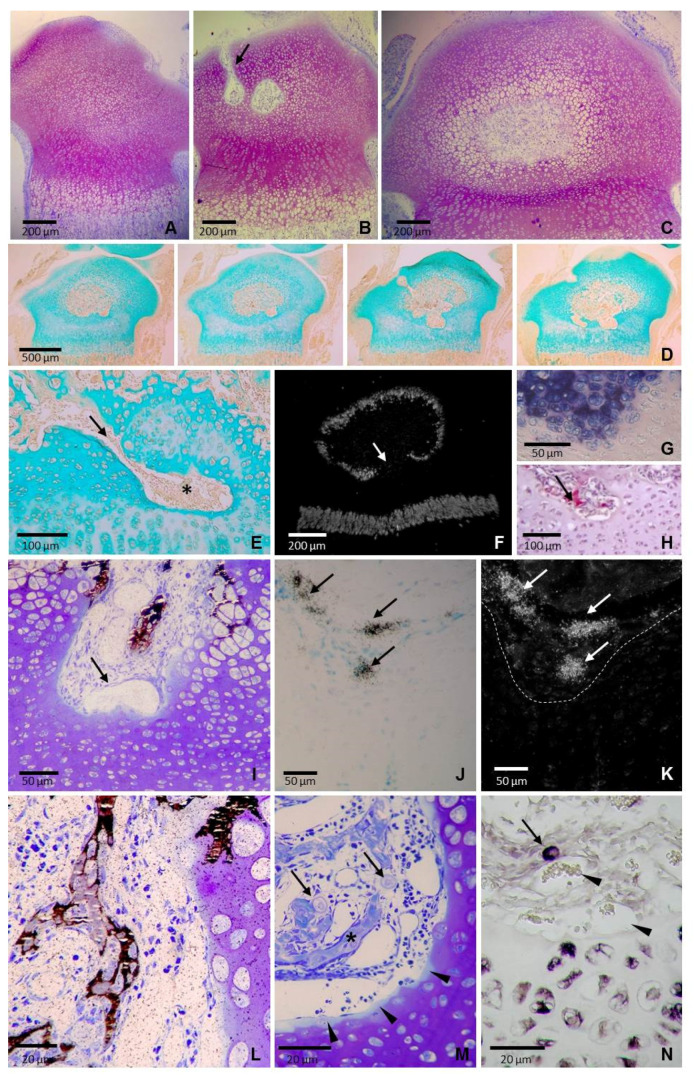
Sections of the tibial epiphysis at the beginning of epiphyseal bone formation. (**A**) Semithin section stained with toluidine of a 3-day-old tibia showing a completely cartilaginous epiphysis. (**B**) Semithin section stained with toluidine of an 8-day-old tibia showing a cartilage canal (arrow) and the early secondary ossification center. (**C**) Semithin section stained with toluidine of an 11-day-old tibia showing a central rounded marrow cavity smoothly bordered by hypertrophic cartilage. (**D**) Series of consecutive paraffin sections of a 14-day-old tibia stained with Alcian blue. The border of the epiphyseal marrow cavity is smooth at the upper and lateral portions and rugged in the lower portion. (**E**) A tongue or finger-like extension of marrow tissue (arrow) deeply penetrating into the epiphyseal cartilage of a 14-day-old tibia (asterisk). (**F**) Dark field microscopy of mRNA expression pattern of type X collagen of a 14-day-old tibia. Expression is found in chondrocytes of both the metaphyseal growth plate and the upper portion of the epiphyseal cartilage but no expression is found in the central region of the lower epiphyseal cartilage (arrow). (**G**) Alkaline phosphatase activity and (**H**) TRAP-positive cells (arrow) in a tongue of epiphyseal marrow tissue penetrating into the epiphyseal cartilage of a 14-day-old tibia. (**I**) A mineralized area adjacent to capillaries in a tongue of epiphyseal marrow tissue of a 14-day-old tibia stained with von Kossa and counterstained with toluidine blue (arrow). (**J**) Bright field and (**K**) dark field images of a section of a 14-day-old tibia hybridized with a ^35^S-labeled riboprobe for osteocalcin. Expressión is observed in some scattered cells (arrows) separated from the border of the cartilage (dotted white line). (**L**) Section of a of a 14-day-old tibia showing von Kossa positive cells located at the mid-depth zone of a finger-like extension of the marrow tissue. Cells are located in the intervascular region, distant from the border with the cartilage. (**M**) Semithin section of a 14-day-old tibia stained with toluidine showing the intimate juxtaposition of the capillaries to the cartilaginous matrix. A thin band of less stained cartilaginous matrix is observed in the border of the cartilage (arrowheads) and two rounded-shaped chondrocyte-like cells with a thin rim of metachromatic substance could be seen (arrows) close to the mineralized area that is stained in blue (asterisk). (**N**) Section of a 14-day-old tibia hybridized with a digoxygenin-labeled riboprobe for type II collagen showing a positive cell being well separated from the border of the cartilage (arrow), close to vascular vessels (arrowheads).

**Figure 2 ijms-22-00900-f002:**
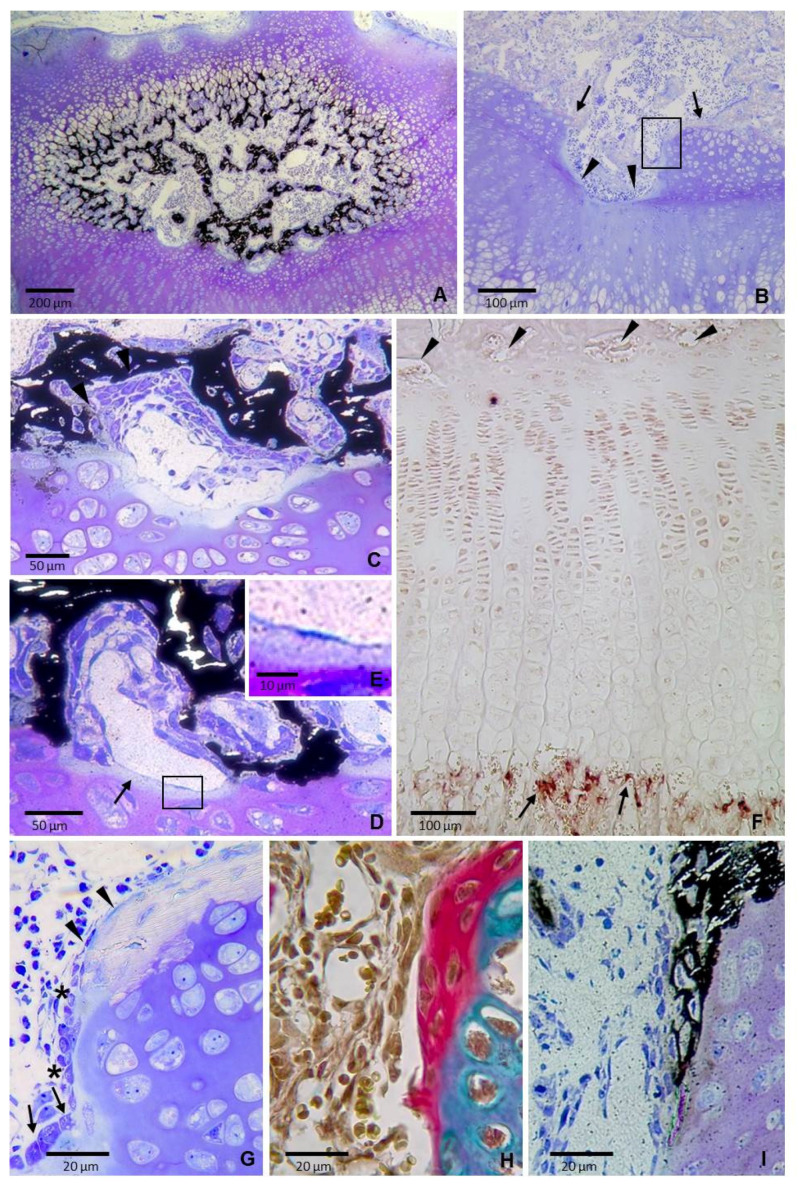
Sections of the tibial epiphysis of 19 day-old rats. (**A**) Section stained with von Kossa and counterstained with toluidine blue showing that the epiphyseal bony tissue displays a “bicycle wheel” pattern at the upper portion and a bridge-like disposition at the lower portion. (**B**) Semithin section stained with toluidine blue showing rounded hollows (arrowheads) and ridges (arrows) in the epiphyseal cartilage. Boxed area is shown at a higher magnification in figure G. (**C**) Section stained with von Kossa and counterstained with toluidine blue showing that the cell composition differs between the upper and lower surfaces of the mineralized plate-like elements. (**D**) Section stained with von Kossa and counterstained with toluidine blue showing a moderately dilated capillary interposed between the cartilage and the bone forming cells. It is close to the surface of the cartilage, where a thin band of less stained cartilaginous matrix is present (arrows). Boxed area is shown at a higher magnification in figure E. (**E**) Magnification of the boxed area in D showing the close contact between the capillary and the cartilage. (**F**) Immunohistochemistry for CD31 showing positive capillaries at the metaphyseal chondro–osseous junction (arrows) whereas virtually no signal is visible in capillaries located at the epiphyseal end (arrowheads). (**G**) Magnification of the boxed area in B showing that the cartilage is directly covered by a layer of mineralized tissue that is weakly stained. Note that chondrocytes are not hypertrophic and have a well-preserved structure. Cells at the surface are flattened (arrowheads) and change to cuboidal at mid-depth of the hollow (arrow) with a transition zone in between (asterisks). (**H**) Section of a region equivalent to that shown in figure G stained with Alcian blue/acid fuchsin. The two tissues are stained differentially, with a sharp boundary visible between them. (**I**) Equivalent region to those shown in figures G and H stained von Kossa and counterstained with toluidine blue showing mineralized matrix being directly deposited on the cartilage.

**Figure 3 ijms-22-00900-f003:**
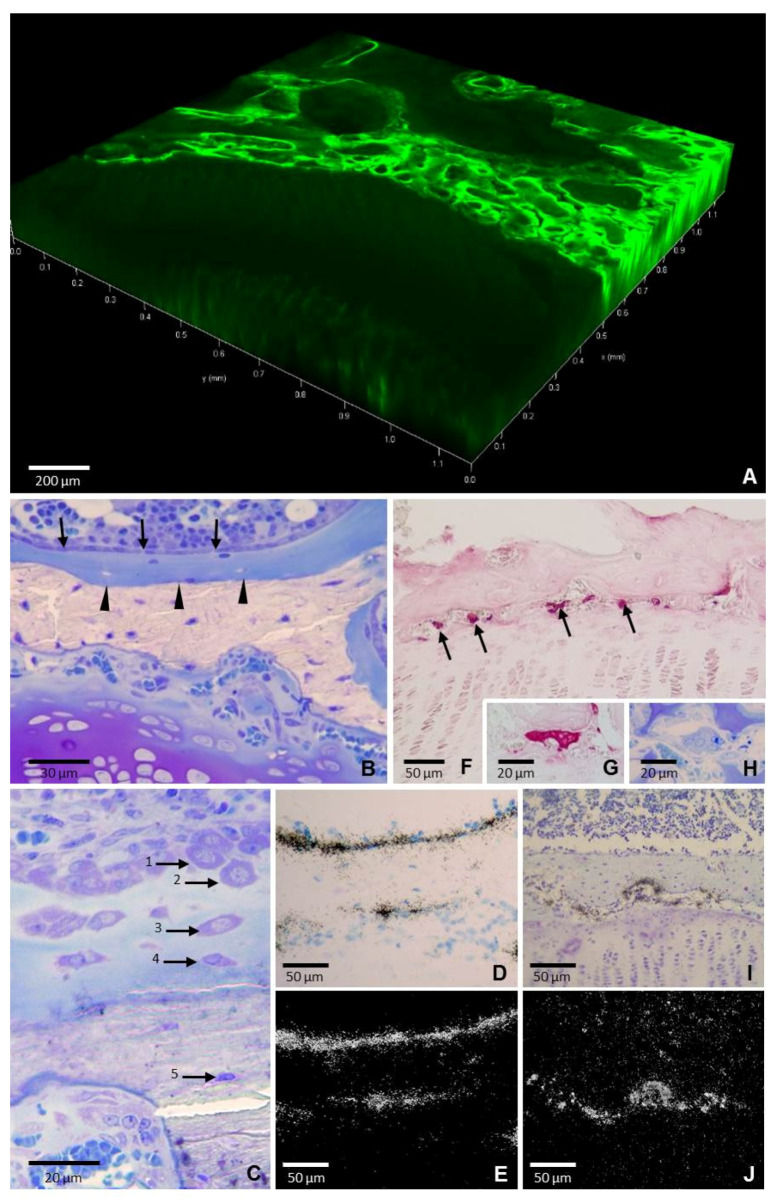
Sections of the tibial epiphysis of a 24-day-old rat. (**A**) Confocal microscopy showing a 3D projection reconstructed from z-stack images of bone tissue labeled with calcein. Fluorescence (green) is located around the blood vessels and this gives rise to an arrangement of circles whereas the underlying growth plate cartilage is completely devoid of fluorescence. (**B**) Semithin section stained with toluidine blue showing a continuous monolayer of cuboidal osteoblasts (arrows) on the surface of the bone plate. Below the osteoblasts is a thick layer of blue-stained osteoid and beneath it, there is the unstained mineralized bone. A sharp boundary between osteoid and mineralized bone is observed (arrowheads). (**C**) Semithin section stained with toluidine blue showing the gradual transformation of the phenotype of the osteoblasts that became trapped in the osteoid (1–2) to osteoid-osteocytes (3–4) and osteocytes (5). (**D**) Bright field and (**E**) dark field images of a section hybridized with a ^35^S-labeled riboprobe for osteocalcin showing a continuous band of positive signal on the surface of the upper side whereas only some scattered cells are positive at the lower side. (**F**) TRAP-positive osteoclasts at the narrow space between the bone plate and the growth plate cartilage (red stained, arrows). (**G**) Higher magnification of a TRAP positive cell. (**H**) Image of the same cell in a semithin section stained with toluidine blue. (**I**) Bright field and (**J**) dark field images of a section hybridized with a 35S-labeled riboprobe for gelatinaseB showing that expression is restricted to the lower side.

**Figure 4 ijms-22-00900-f004:**
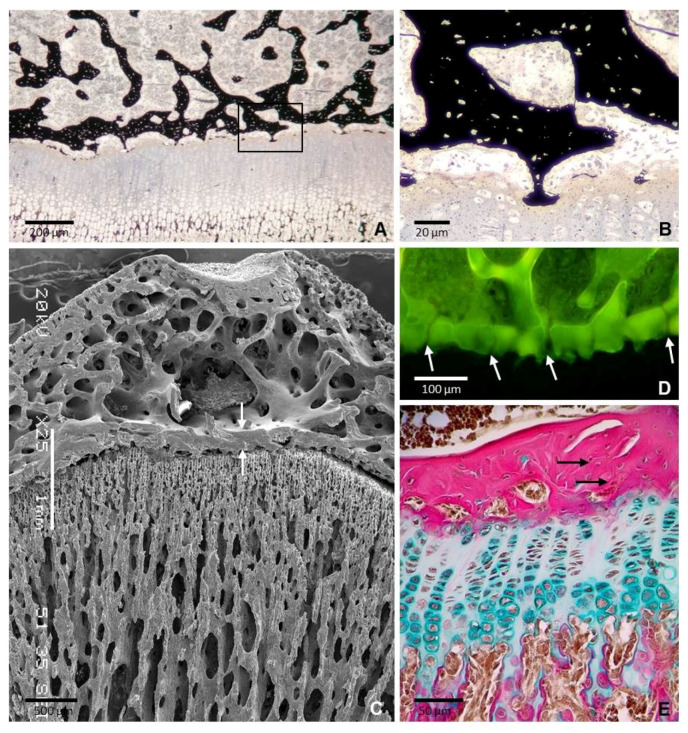
Sections of the tibial epiphysis of a 35 day-old rat. (**A**) Section stained with von Kossa showing that the epiphyseal bone plate and the underlying growth plate are bridged by a number of thin bony extensions. Boxed area is shown at a higher magnification in figure B. (**B**) Magnification of the boxed area in A showing a bone-cartilage contact. Note that osteocytes in the mineralized matrix are not aligned do not show a clear orientation. (**C**) SEM image showing that the epiphyseal plate is thicker than the cortical bone of the epiphysis (arrows). Note the continuity of the bone plate with the trabecular meshwork of the epiphysis. (**D**) Confocal microscopy image of a 3D projection reconstructed from z-stack images showing the foramina of the plate through which blood vessels passed (arrows). (**E**) Paraffin section stained with Alcian blue/acid fuchsin showing that collagen fibers are densely packed but are not arranged in a regular parallel pattern (arrows).

## Data Availability

The data that support the findings of this study are available from the corresponding author upon reasonable request.

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
