# Peer review of "The Formation of the Epiphyseal Bone Plate Occurs via Combined Endochondral and Intramembranous-Like Ossification"

_ijms, 2021, doi:10.3390/ijms22020900_

Round 1

Reviewer 1 Report

  The Article “The formation of the epiphyseal bone plate occurs via combined endochondral and intramembranous ossification” describes that the epiphyseal bone plate is generated through a endochondral and intramembranous ossification processes.  This manuscript contains sufficient interest and originality to merit publication.  However, I have several concerns with this manuscript.  Please confirm my comments as indicated below.

  1. Authors showed RNA ISH about Type X collagen using mouse RNA probe in Fig.1C. Please show the gene homology (%) of this probe for rat gene in the Materials and methods section.  Mouse probe sequence is 100% homology with Rat sequence?

  1. Figure 1H: can the authors show the distinct chondrogenic cell feature of rounded-shaped cells (indicated by arrows) by, e.g. chondrocyte marker IHC?

  1. Figure 2D: can the authors show the endothelial cell feature of capillary structures (indicated by arrows) by, e.g. vascular marker IHC?

  1. Figure 3B: can the authors show the osteogenic feature of monolayer cells on the surface of the bone plate (indicated by arrows and arrowheads) by, e.g. osteoblast marker IHC?

  1. Line367: “the structural characteristics of these capillaries do not match with those of the metaphyseal type H vessels.” could be shown by immunohistochemical staining analysis for a marker (CD31 and Endomucin) of type H vessels.

  1. A diagram could be made to illustrate the novelty of this paper in a visually understandable state.

Author Response

R: The Article “The formation of the epiphyseal bone plate occurs via combined endochondral and intramembranous ossification” describes that the epiphyseal bone plate is generated through a endochondral and intramembranous ossification processes.  This manuscript contains sufficient interest and originality to merit publication.  However, I have several concerns with this manuscript.  Please confirm my comments as indicated below.

A: We appreciate very much the Referee recognizing that our manuscript contains sufficient interest and originality to merit publication. He/she raises important issues and his/her inputs were very helpful for improving the manuscript. We agree with almost all his/her comments and we have revised our manuscript accordingly. We respond below to each of the reviewer’s comments

Authors showed RNA ISH about Type X collagen using mouse RNA probe in Fig.1C. Please show the gene homology (%) of this probe for rat gene in the Materials and methods section.  Mouse probe sequence is 100% homology with Rat sequence?

We have taken this remark into account and included in the revised manuscript the percentage of sequence similarity of the rat and mouse genes.

Figure 1H: can the authors show the distinct chondrogenic cell feature of rounded-shaped cells (indicated by arrows) by, e.g. chondrocyte marker IHC?

We have taken this remark into account and included in the revised manuscript an in situ analysis of expression of type-II collagen to characterize the chondrocyte-like cells found in intervascular space, freed from the cartilage matrix.

Figure 2D: can the authors show the endothelial cell feature of capillary structures (indicated by arrows) by, e.g. vascular marker IHC?

We think that the method used, resin-embedded semithin sections, provides a good structural preservation that allows a clear identification of the endothelial cells of the capillaries. Nevertheless, we have included an enlarged inset in revised figure for better illustration of the close contact between the endothelial cell and the cartilage matrix. In addition, as explained after the next comment below, we have included in the revised manuscript an immunohistochemical analysis of CD31.

Figure 3B: can the authors show the osteogenic feature of monolayer cells on the surface of the bone plate (indicated by arrows and arrowheads) by, e.g. osteoblast marker IHC?

We have taken this remark into account and included in the revised manuscript an in situ analysis of expression of osteocalcin, a specific bone protein produced exclusively by osteoblasts that is widely considered a marker of metabolic activity of these cells.

Line367: “the structural characteristics of these capillaries do not match with those of the metaphyseal type H vessels.” could be shown by immunohistochemical staining analysis for a marker (CD31 and Endomucin) of type H vessels.

 We have taken this remark into account and included in the revised manuscript an immunohistochemical analysis of CD31 to analyze its expression in epiphyseal capillaries and compare it with that of metaphyseal capillaries.

A diagram could be made to illustrate the novelty of this paper in a visually understandable state.

In the revised manuscript we have rewritten part of the Discussion section to highlight that our manuscript provides probably the first systematic analysis of the formation of the epiphyseal bone plate and that our study demonstrates that this process involves a mode of ossification that is temporally, spatially and histologically distinct from other bony structures.

Reviewer 2 Report

Fernandez-Iglesias and collaborators present detailed histologic analyses of the formation of the epiphyseal bone plate of the proximal tibia, during endochondral bone formation, in postnatal rats (P14, P19, P24, P35). This work is interesting since most of the observations published on bone formation have been made looking at the primary ossification center, in the metaphyseal region. However, several points should be considered to improved the quality and the overall interest of this manuscript:

  • The written descriptions of the figures are often hard to follow. Schematic representations of the most important aspects of the phenotypes described could help significantly the reader to understand what the authors are talking about, with the precise localization and orientation of the aspects discussed. In figures, low magnification images (of the epiphysis of the tibia) with boxed areas shown at higher magnification would also help the readers to know the exact localization of the high-magnification images.
  • A large amount of the phenotypic description of the tibial epiphysis is not shown. This is the case at the beginning of the result section with description of P3, P5 and P8 epiphyses without any associated figures. Also, the temporal evolution described in the text is hard to picture (for example “the secondary ossification center was first spherical and became increasingly irregular”) solely based on the images provided, in part because the bone and cartilage phenotypes in the epiphysis change rather quickly between the different stages analyzed (in particular between P14 and P19). It would be much better to provide in the same figure images of cross-sections of the epiphyses at different stages (adding more stages) all stained the same way (alcian blue/acid fuchsin or H&E). In situ hybridizations with specific markers (Col2, ColX, OPN and Col1) should be performed as well on the adjacent sections, to better visualize the evolution of cell differentiation overtime.
  • It would be interesting and important to provide additional immunostaining to clearly visualize blood vessels (CD31 and/or Endomucin), and osteoprogenitor cells (OSX). Ideally, lineage-tracing strategies could be used (in mice with the appropriate genetic tools) to better visualize the progression of osteoprogenitors, and blood vessels during the formation of the secondary ossification center. Lineage tracing could also be used to determine if contrary to what happens during the formation of the primary ossification center, hypertrophic chondrocytes do or do not contribute to the formation of the secondary ossification center.
  • The notion that the formation of the epiphyseal bone plate occurs via combined endochondral and intramembranous ossification (the title of this manuscript) appears somewhat subjective, and probably misleading. Intramembranous ossification occurs without cartilage. The formation of the secondary ossification center in tibial epiphyses would not occur without cartilage. The title should therefore be modified accordingly. The term “intramembranous-like ossification” could be used, but should be better explained and must be justified in the main text, so that the readers really understand what the authors talk about.
  • The authors mention that they provide for the first time an accurate description of the events involved in the formation of the epiphyseal bone plate and disclose peculiarities of physiological interest. They could better explain what are the physiological interests revealed by their data in the discussion section.

Author Response

R: Fernandez-Iglesias and collaborators present detailed histologic analyses of the formation of the epiphyseal bone plate of the proximal tibia, during endochondral bone formation, in postnatal rats (P14, P19, P24, P35). This work is interesting since most of the observations published on bone formation have been made looking at the primary ossification center, in the metaphyseal region. However, several points should be considered to improved the quality and the overall interest of this manuscript:

A: We thank very much the referee for his/her overall positive evaluation of our manuscript and also for his/her helpful comments.  We agree and accept essentially all of his/her suggestions in the revised version of the manuscript. We respond below to each of the reviewer’s comments

The written descriptions of the figures are often hard to follow. Schematic representations of the most important aspects of the phenotypes described could help significantly the reader to understand what the authors are talking about, with the precise localization and orientation of the aspects discussed. In figures, low magnification images (of the epiphysis of the tibia) with boxed areas shown at higher magnification would also help the readers to know the exact localization of the high-magnification images.

We agree with the reviewer that descriptions of the cellular changes could be rewritten in a more schematic form to make easier for the reader to follow. In the revised manuscript we have we have changed the way of presenting the results by dividing the formation of the bone plate  into four stages characterized by changes in the amount of cartilage and bone formation. We have also rewritten most of the Results section to clarify the relevant aspects of the cellular changes occurring during the formation of the epiphyseal bone plate. We have also rewritten some of the descriptions in the legends of figures and delineated with boxes in the low magnification images the areas in them shown at higher magnification.

A large amount of the phenotypic description of the tibial epiphysis is not shown. This is the case at the beginning of the result section with description of P3, P5 and P8 epiphyses without any associated figures. Also, the temporal evolution described in the text is hard to picture (for example “the secondary ossification center was first spherical and became increasingly irregular”) solely based on the images provided, in part because the bone and cartilage phenotypes in the epiphysis change rather quickly between the different stages analyzed (in particular between P14 and P19). It would be much better to provide in the same figure images of cross-sections of the epiphyses at different stages (adding more stages) all stained the same way (alcian blue/acid fuchsin or H&E). In situ hybridizations with specific markers (Col2, ColX, OPN and Col1) should be performed as well on the adjacent sections, to better visualize the evolution of cell differentiation overtime.

We agree that description of the findings must be supported by representative figures but it is also true that the number of figures in an article is not unlimited. We thought that the 31 microphotographs included in the first version of manuscript supported the main body of the paper but it is also true that some descriptions were not accompanied by figures. Then, we have added 10 new figures, including representative figures of P3, P8 and P11 that were not in the previous version. Thus, the revised version of our manuscript contains a total of 41 microphotographs. Rearrangement of the micrographs in the plates with the aim of having equivalent areas at different stages in the same plate was probed. However, after several trials we reached the conclusion that the arrangement of the microphotographs by developmental stages gave an easily interpreted visualization of the formation process.

It would be interesting and important to provide additional immunostaining to clearly visualize blood vessels (CD31 and/or Endomucin), and osteoprogenitor cells (OSX). Ideally, lineage-tracing strategies could be used (in mice with the appropriate genetic tools) to better visualize the progression of osteoprogenitors, and blood vessels during the formation of the secondary ossification center. Lineage tracing could also be used to determine if contrary to what happens during the formation of the primary ossification center, hypertrophic chondrocytes do or do not contribute to the formation of the secondary ossification center.

We have taken this remark into account and included in the revised manuscript an in situ analysis of expression of osteocalcin, a specific bone protein produced exclusively by osteoblasts, and also an immunohistochemical analysis of CD31 to analyze its expression in epiphyseal capillaries and compare it with that of metaphyseal capillaries. We agree that lineage-tracing strategies would be the most powerful tools to understand the progression of osteoprogenitors, and blood vessels during the formation of the secondary ossification center. However, they are fairly complicated and time consuming and it is impossible for us to incorporate them into the present work.  Nevertheless, we think that our present study gives an overall overview of the whole process and provides a framework for future research on this subject.

The notion that the formation of the epiphyseal bone plate occurs via combined endochondral and intramembranous ossification (the title of this manuscript) appears somewhat subjective, and probably misleading. Intramembranous ossification occurs without cartilage. The formation of the secondary ossification center in tibial epiphyses would not occur without cartilage. The title should therefore be modified accordingly. The term “intramembranous-like ossification” could be used, but should be better explained and must be justified in the main text, so that the readers really understand what the authors talk about.

We have taken this comment into account and included the term “intramembranous-like ossification” in the revised manuscript. Furthermore, we have rewritten part of the Discussion section in order to explain the contribution of our work.

The authors mention that they provide for the first time an accurate description of the events involved in the formation of the epiphyseal bone plate and disclose peculiarities of physiological interest. They could better explain what are the physiological interests revealed by their data in the discussion section.

We have taken this comment into account and have added to the Discussion section a paragraph with the possible physiological implications of the results obtained.

Reviewer 3 Report

Authors analyzed a poorly investigated field. The methodology is good and they show really nice pictures.

The study could be interesting if implemented with some pathological condition showing a defect of epiphyseal region and a defect of one or both the ossification type. I would also suggest to stain samples with some bone markers involved in the ossification process to describe which pathway could be involved and to show if there’s an imbalance of both or a collaboration between them to create new bone.

I suggest to insert the dimension of the scale bar in the pictures and not in the legend to be more impressive for readers.

In picture 1 you show dark filed. It is possible to show mrna colocalization in bone cells?

Author Response

Authors analyzed a poorly investigated field. The methodology is good and they show really nice pictures.

We appreciate the Referee recognizing that our manuscript is focused on a poorly investigated field and thank him/her very much for the positive assessment of the methodology and the pictures of our work. We agree and accept essentially all of his/her suggestions in the revised version of the manuscript. We respond below to each of the reviewer’s comments

The study could be interesting if implemented with some pathological condition showing a defect of epiphyseal region and a defect of one or both the ossification type. I would also suggest to stain samples with some bone markers involved in the ossification process to describe which pathway could be involved and to show if there’s an imbalance of both or a collaboration between them to create new bone.

We agree that it is very interesting to study pathological conditions that could affect the normal formation of the epiphyseal bone tissue. In fact we work in our laboratory with animal models for the study of two human diseases that cause growth retardation and bone deformities: X-linked hypophosphatemic (XLH) rickets (doi: 10.1016/j.bone.2018.08.004) and chronic kidney disease (CKD)  (doi.org/10.3390/ijms21124519). Our intention for a future study is to analyze if the structure of the epiphyseal growth plate is affected in such pathological conditions. But we have not started with this study and it is impossible to obtain this data in a short time to incorporate into the present work.  The main goal of the present research is to give an overall overview of the whole process that could provide a framework for future research on this poorly investigated field.

I suggest to insert the dimension of the scale bar in the pictures and not in the legend to be more impressive for readers.

We have taken this remark into account and included in the revised manuscript the scale bars are inserted in the pictures and not in the legends

In picture 1 you show dark filed. It is possible to show mrna colocalization in bone cells?

We have taken this remark into account and microphotographs of osteocalcin and type II collagen ISH have been added to the Picture 1.

Round 2

Reviewer 1 Report

Revised maniscript is an " excellent", I have no hesitation in recommending it for publication.

Reviewer 2 Report

The authors have satisfactorily addressed all the points raised to improve the quality and interest of their manuscript.